# Watching the Saltmarsh Grow: A High-Resolution Remote Sensing Approach to Quantify the Effects of Wetland Restoration

Ashley J. Rummell *, Javier X. Leon, Hayden P. Borland, Brittany B. Elliott, Ben L. Gilby, Christopher J. Henderson and Andrew D. Olds

School of Science, Technology and Engineering, University of the Sunshine Coast, Maroochydore DC, QLD 4558, Australia
* Correspondence: arummell@usc.edu.au

**Abstract:** Coastal wetlands are restored to regenerate lost ecosystem services. Accurate and frequent representations of the distribution and area of coastal wetland communities are critical for evaluating restoration success. Typically, such data are acquired through laborious, intensive and expensive field surveys or traditional remote sensing methods that can be erroneous. Recent advances in remote sensing techniques such as high-resolution sensors (<2 m resolution), object-based image analysis and shallow learning classifiers provide promising alternatives but have rarely been applied in a restoration context. We measured the changes to wetland communities at a 200 ha restoring coastal wetland in eastern Australia, using remotely sensed Worldview-2 imagery, object-based image analysis and random forest classification. Our approach used structural rasters (digital elevation and canopy height models) and a multi-temporal technique to distinguish between spectrally similar land cover. The accuracy of our land cover maps was high, with overall accuracies ranging between 91 and 95%, and this supported early detection of increases in the area of key ecosystems, including mixed she-oak and paperbark (10 ha), mangroves (0.91 ha) and saltmarsh (4.31 ha), over a 5-year monitoring period. Our approach provides coastal managers with an accurate and frequent method for quantifying early responses of coastal wetlands to restoration, which is essential for informing adaptive management in the regeneration of ecosystem services.

**Keywords:** coastal wetland; restoration; Worldview-2; object-based image analysis; change detection; multi-temporal

## 1. Introduction

Large areas of coastal wetlands have been degraded or lost globally due to the expansion of agriculture and urbanization. This has depleted valuable ecosystem services, including fisheries habitat, carbon storage, nutrient cycling and coastline protection [1–3]. Remnant coastal wetlands and the ecosystem services they provide are also threatened by the future impacts of climate change and sea-level rise such as coastal squeeze effects and saltmarsh encroachment by mangroves [4,5]. Coastal wetland communities (e.g., mangroves and saltmarshes) are increasingly targeted in critical restoration projects seeking to promote these lost ecosystem services [6,7]. Coastal wetland communities are restored, for example, via passive techniques that promote natural recovery through the removal of tidal barriers to re-establish tidal connectivity or by active techniques such as propagule planting or re-engineering [8]. Restoration projects are, however, not always successful because the responses of coastal wetland communities can be dynamic, due to fluctuations in tidal pulses, flushing, rainfall and restoration strategies undertaken and by long-term threats, including sea-level rise [9–11]. For example, when hydrological connections are impaired, flushing during flood events can be poor, resulting in damage or dieback of restored sites [12]. Additionally, monitoring for degradation events in wetland ecosystems is also vital, as this can identify areas that can be improved with restoration actions [13–15].

Frequent monitoring of changes in the area and distribution of coastal wetland communities at restoration sites over time is required to inform adaptive management approaches that increase the likelihood of restoration success, but this can be both time consuming and expensive [16].

Assessments on the degradation or regeneration of ecosystem services are desirable for stakeholders and coastal managers because they identify areas that could benefit from targeted ecosystem repair, inform the effectiveness of restoration actions and estimate financial incentives as part of blue carbon accounting schemes and offsets [17,18]. Identifying changes to key ecosystem services (e.g., fisheries habitat and carbon sequestration) relies on robust datasets that quantify changes in the type, area and distribution of coastal wetland communities after restoration actions [19,20]. These datasets have historically been quantified via intensive field-based methods that precisely measure the structure, condition and diversity of coastal wetland communities. These approaches are, however, often limited by their spatial and temporal coverages [21,22]. Field-based surveys in coastal wetlands are also impractical because they are laborious, disruptive to plant communities and often restricted by accessibility [23]. The need to frequently measure restoration effects at larger spatial extents is, however, escalating with increases in the number and scale of restoration projects worldwide. It is for this reason that remote sensing approaches are becoming more favored over field-based monitoring in most settings [8,24].

Traditionally, remote sensing approaches have measured the distribution and coverage of estuarine ecosystems using low- to medium-resolution (10 to 30 m) satellite-borne sensors (e.g., Landsat, Sentinel, and SPOT), pixel-based image analysis and parametric classifiers (e.g., maximum likelihood classification) [25–27]. These methods are used to measure changes to communities over large spatial extents and are cost-effective but do not provide high accuracies (60 to 80%) or detailed changes to small-habitat patches [28,29]. Some low-resolution sensors (e.g., Landsat-8 and Sentinel-1 or 2) are being used in integrated databases and operational systems to improve data sharing and categories of temporal change to wetland communities, which can better inform stakeholders and multi-criteria analysis [30–32]. Technological advancements in the last decade have improved the capability of image acquisition platforms, sensors and image analysis techniques to detect change [33]. For example, high-spatial-resolution (0.5–5 m), multi-spectral sensors such as Worldview-2 provide more detailed imagery to differentiate between distinct coastal wetland land cover classes [34,35]. Image analysis techniques have also improved from simple pixel-based approaches to robust object-based image analyses, which yield higher overall accuracies when classifying estuarine land cover [36–38]. The development of more advanced classification algorithms has enhanced the accuracy of estuarine land cover mapping, with progressions from simplistic parametric classifiers such as maximum likelihood classifiers to non-parametric machine learning and deep learning classifiers [39]. Deep learning classifiers (e.g., convolution neural networks and recurrent neural networks) can deliver high accuracy results but are computationally expensive and require large training datasets [40,41]. Shallow learning methods (e.g., random forest and support vector machines) can deliver high classification accuracies, which rival deep learning classifiers whilst also requiring fewer training samples [42,43].

Studies in estuarine environments have successfully paired high-resolution sensors (e.g., Worldview-2) with object-based image analysis and non-parametric classifiers (e.g., random forests and support vector machines) and recorded accuracies exceeding 90% for coastal wetland communities and 80–90% for species-level identifications (e.g., within mangrove forests) [39,44,45]. These approaches, however, often classify estuarine communities from a singular time point because generating multi-temporal maps and detecting changes to these communities over time is difficult due to the dynamic nature of tides, rainfall and seasons affecting the spectral and structural values of classes [46,47]. Restoration likely intensifies variations in the spectral and structural values of vegetation classes due to the introduction of volatile environmental conditions causing dieback events, and therefore,

a high-resolution, accurate and frequent approach to detecting the response of coastal wetland communities to restoration actions has not been identified [48,49].

The aim of this study was to monitor the response of coastal wetland communities to restoration actions at a high temporal frequency using an approach that features high-resolution Worldview-2 imagery, object-based image analysis and random forest classification. We determined the viability of our approach by quantifying the overall accuracy of each land cover map, the users' and producers' accuracy of each class, and the trends and change in area of land cover classes in response to restoration actions. Our approach can be used by coastal managers to monitor the success of restoration actions of coastal wetland communities, inform adaptive management strategies and evaluate the regeneration of ecosystem services.

## 2. Materials and Methods

### 2.1. Study Area

Surveys were conducted at Yandina Creek wetlands in eastern Australia (Figure 1). The Yandina Creek wetlands are located in the lower Maroochy River catchment, approximately 15 km upstream from the river mouth and experience semi-diurnal tides with the highest astronomical tides of 1.41 m [50] and an annual rainfall of 1483.4 mm [51]. The Maroochy River catchment is dominated by coastal wetland complexes characterized by both estuarine and palustrine communities, including river (*Aegiceras corniculatum*), grey (*Avicennia marina*), orange (*Bruguiera gymnorhiza*), red (*Rhizophora stylosa*) and milky (*Excoecaria agallocha*) mangroves, saltwater couch (*Sporobolus virginicus*), streaked arrow grass (*Triglochin striata*) and sea blight (*Suaeda australis*) saltmarshes, swamp she-oak (*Casuarina glauca*), broad-leaved paperbark (*Melaleuca quinquenervia*), and common reed (*Phragmites australis*). The Maroochy River catchment has experienced significant land clearing and modification since the 1950s, with the expansion of agriculture and urban development leading to a 30% reduction in the area of mangroves between 1988 and 2016 [52].

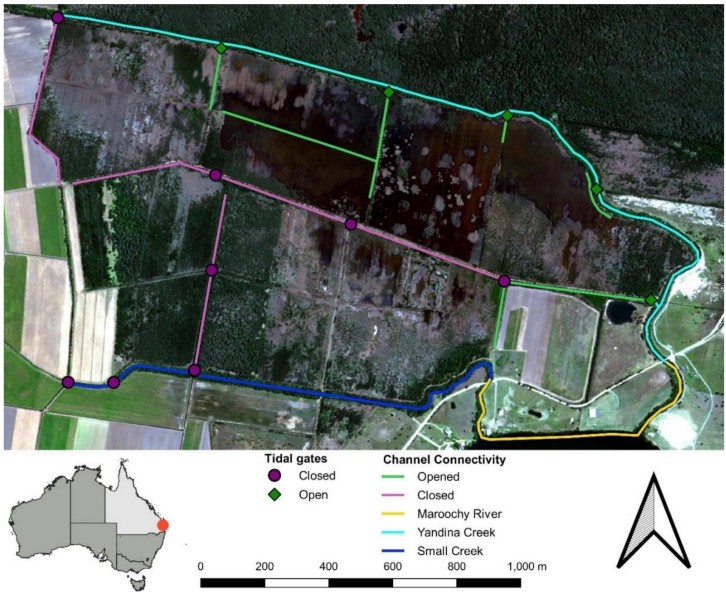

**Figure 1.** The Yandina Creek wetlands, and the distribution of tidal gates, drainage channels and natural waterways on the site.

The Yandina Creek wetlands restoration site covers approximately 200 ha and is traversed by Yandina and Small Creeks. The wetland was disconnected from the downstream Maroochy River catchment by 13 tidal gates and has supported sugar cane and cattle farming since the 1950s. Tidal flow was restored to the site in May 2018, when five tidal gates were removed in the northern, central and eastern regions of the site (Figure 1). Mangrove and saltmarsh plants are colonizing these areas, and it is hoped that the re-

covery of these communities will promote a diversity of ecosystem services, including denitrification, carbon sequestration, biodiversity, and habitat condition and coverage. The southern and western regions of the site remain closed to tidal inundation to promote recovery of palustrine vegetation communities, which are dominated by swamp she-oak and broad-leaved paperbark.

### 2.2. Land Cover Classification and Data Acquisition

Land cover at Yandina Creek wetlands was described via nine classes (Table 1), which represented either regional ecosystems (e.g., mixed she-oak and paperbark, mangroves, saltmarsh, mixed shrubs and grasses), dense stands of individual plant species (e.g., mangrove fern, common reed), tidal waters, or exposed soils. The regional ecosystems framework is a fine-scale classification scheme for describing vegetation communities across Queensland, Australia, based on their bioregion, vegetation structure, composition, hydrological exposure (i.e., tidal or water table inundation) and geology (i.e., terrain and substrate) [53,54]. This framework was used to classify mixed vegetation communities at Yandina Creek wetlands because these communities are typically heterogenous and are closely aligned with the characteristics of some regional ecosystem types (e.g., RE: 12.1.3 a, b, c, 12.1.2, 12.3.20, 12.3.8).

**Table 1.** Summary of the nine classes used to characterize land cover types at Yandina Creek wetlands, their regional ecosystem classification and the dominant species or features within each class.

| Class | Regional Ecosystem | Dominant Species or Feature |
|---|---|---|
| Mangroves | 12.1.3 a, b, c | *Aegiceras corniculatum, Avicennia marina, Bruguiera gymnorhiza* |
| Mangrove fern | - | *Acrostichum speciosum* |
| Saltmarsh | 12.1.2 | *Sporobolus virginicus, Bacopa monnieri, Cycnogeton striata* |
| Mixed she-oak and paper bark | 12.1.1 | *Melaleuca quinquenervia, Casuarina glauca +/− Eucalyptus tereticornis +/− Hibiscus tiliaceus* |
| Mixed shrubs and grasses | 12.3.8 | *Cyperus polystachyos, Leersia hexandra, Juncus kraussi, Lomandra hystix* |
| Common reed | - | *Phragmites australis +/− Baccharis halimifolia* |
| Vegetation dieback | - | Dying vegetation |
| Exposed soil | - | Mud, dirt, rock, road |
| Tidal waters | - | Tidal inundation, ponding, creeks, drainage channels |

Between 1500 and 3000 training and validation samples (Table 2) were randomly generated to describe the nine land cover classes (~100–300 samples per class) in a combination of field surveys and, when areas were inaccessible, very high-resolution (~7 cm) Nearmap imagery [55] and drone orthomosaics (<2 cm) derived from a Phantom-4 RTK drone (PT4) (2 cm) (Table S1). We adopted a 70/30 training to testing sample ratio, which is commonly applied in remote sensing classification and is suggested by recent research [56]. Additionally, we also followed a simple random design in accordance with Stehman and Foody [57].

Field surveys were conducted across the six vegetation classes at three different locations (n = 20) in Autumn 2020. These surveys were conducted in regions of Yandina Creek wetlands that experienced little change over the period since restoration commenced to ensure these samples remained consistent throughout the study. These surveys identified community composition and structure with 10 m$^2$ quadrats, and the middle of each site

was recorded with a CHC X91+ Real Time Kinematic GPS, which has a horizontal and vertical precision of 8 mm and 15 mm, respectively. The PT4 drone was also flown over two regions that were difficult to access in a stratified pattern at 60 m altitude and image overlapping was set to 85% [58]. The PT4 is an ideal alternative to field validation, as it records highly accurate (<1 cm horizontal and <2 cm vertical) georeferenced imagery at a very fine resolution (<2 cm) [58]. The majority of Yandina Creek wetlands falls within 5.5 km$^2$ of a class "C" airspace (CASA) (i.e., no-fly zone), which limited coverage and frequency of drone surveys. Structure from motion (SfM) was then used to generate two very high-resolution (<2 cm) orthomosaics of regions at Yandina Creek wetlands that were difficult to access covering areas 30 and 12.5 ha [59]. Briefly, SfM is a method of photogrammetry that matches common points in overlapping two-dimensional images to generate a three-dimensional point cloud [60].

**Table 2.** Summary of training samples, validation samples and overall accuracy for each corresponding land cover map.

| Map Date | Sampling Event | Time | Training Samples | Validation Samples | Overall Accuracy |
|---|---|---|---|---|---|
| 23 July 2017 | 2017 spring | 0 months | 1119 | 449 | 91% |
| 24 April 2018 | 2018 autumn | 6 months | 1309 | 508 | 92% |
| 2 August 2018 | 2018 spring | 12 months | 2067 | 779 | 94% |
| 27 April 2019 | 2019 autumn | 18 months | 1156 | 496 | 91% |
| 18 September 2019 | 2019 spring | 24 months | 1808 | 679 | 95% |
| 20 March 2020 | 2020 autumn | 30 months | 1128 | 468 | 92% |
| 26 September 2020 | 2020 spring | 36 months | 1338 | 548 | 92% |
| 23 April 2021 | 2021 autumn | 42 months | 1458 | 618 | 94% |
| 25 September 2021 | 2021 spring | 48 months | 1712 | 651 | 93% |

### 2.3. Data Processing

All land cover maps were produced by combining several steps, including pre-processing, object-based image analysis and random forest classification (Figure 2). Worldview-2 satellite imagery was selected to inform estimates of land cover classes at each site because it offers a range of multispectral bands (Table S2). Nine total Worldview-2 images were acquired from Digital Globe, one prior to restoration commencing, and then eight throughout the restoration process following a bi-sampling approach (Table S1).

Images were pre-processed via radiometric calibration with parameters provided by Digital Globe to convert digital numbers to the top of atmosphere reflectance values. Each Worldview-2 image was also pan-sharpened with the nearest neighbor diffusion filter (NNDiffuse), to increase spatial resolution from 2 m to 0.5 m by fusing the multispectral bands to the corresponding panchromatic band [61]. We opted to pan-sharpen the Worldview-2 imagery because previous research indicates greater overall accuracies when mapping coastal wetland communities than those at the standard 2 m resolution [42]. The NNDiffuse algorithm is a recommended technique for image fusion with Worldview imagery because it maintains spectral and textural information with small errors (~5%) [61].

We generated three additional spectral indices, including normalized difference vegetation index (NDVI), red-edge simple ratio (RE-SR) and Worldview-water index (WV-WI) (Table 3). These spectral indices were selected because they provide a wider range of information than multispectral bands do independently. For example, NDVI ratios are common amongst vegetation mapping applications because they are effective at representing high moisture absorption and chlorophyll in vegetation communities across a variety of conditions [42,62,63]. Similarly, the RE-SR is a valuable index featured by Worldview sensors that are used to distinguish between mangrove species and other vegetation classes by identifying variation in chlorophyll counts and moisture retainment [64,65]. The WV-WI is

specific to Worldview sensors and was used to highlight water and shadow pixels because it provides the greatest ability to distinguish between water, soil and vegetation classes [66].

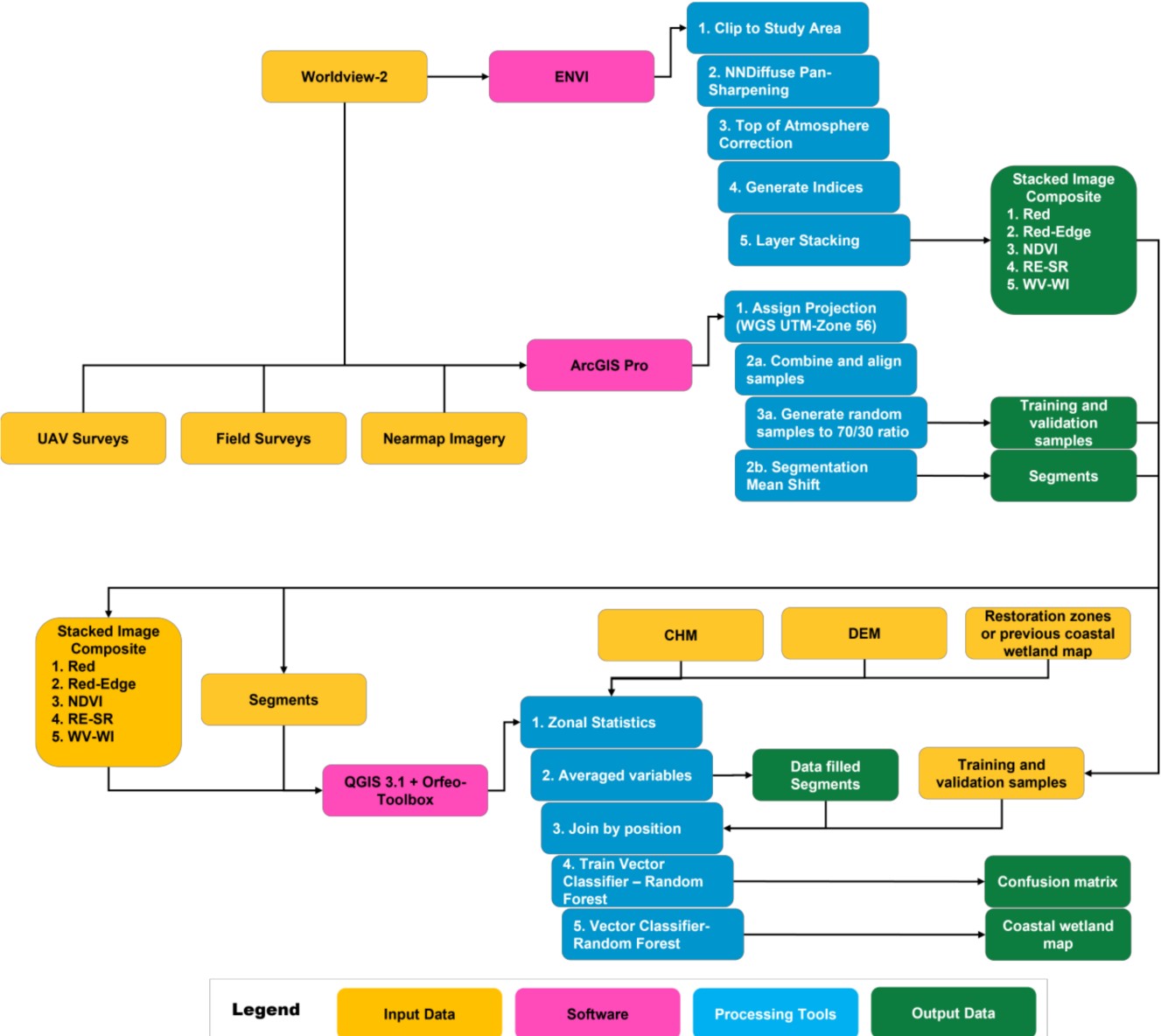

**Figure 2.** Methods workflow summarizing the combination of datasets, software packages and approaches used to classify land cover types for each map.

A canopy height model (CHM) and digital elevation model (DEM) were used to discriminate between spectrally similar mixed she-oak and paperbark, and mangrove classes. Both the CHM and DEM were derived from LiDAR point clouds captured by Sunshine Coast Regional Council in Autumn 2018 [67]. The CHM and DEM have a resolution of 1 m each, and the CHM was created by differing a digital surface model (DSM) from the DEM. We generated a "restoration zone" raster based on dominant groupings of vegetation communities and exposure to different restoration treatments, drainage channels and tidal gates. Following a multi-temporal approach, we used the prior land cover map as a variable in the classification of each successive map, as this technique maximizes image replicability over short time scales [68]. A total of nine datasets, including two pan-sharpened multispectral images (red and red-edge), three spectral indices (NDVI, RE-SR, WV-WI), three spatial rasters (CHM, DEM, restoration zones) and the previous land cover map, were combined to form a stacked image (Table 3).

**Table 3.** Summary of the image layers used to classify the nine land cover classes at Yandina Creek wetlands.

| Data Source | Variable | Variable Specifications |
|---|---|---|
| Worldview-2 spectral indices | Red | 624–694 |
| | Red-Edge | 99–749 |
| | Normalized Difference Vegetation Index | $\text{NDVI} = \frac{(\text{NIR1} - \text{Red})}{(\text{NIR1} + \text{Red})}$ |
| | Red-Edge Simple Ratio | $\text{RE} - \text{SR} = \frac{(\text{RE})}{(\text{Red})}$ |
| | Worldview-Water Index | $\text{WV} - \text{WI} = \frac{(\text{CB} - \text{NIR2})}{(\text{CB} + \text{NIR2})}$ |
| LiDAR | Canopy Height Model (CHM) | $\text{CHM} = \text{DSM} - \text{DEM}$ |
| | Digital Elevation Model (DEM) | - |
| | Digital Surface Model (DSM) | |
| Raster | Restoration zones | - |
| | Previous land cover map | - |

### 2.4. Object-Based Image Analysis

We classified land cover for each time point by using independent training points and applying an object-based image analysis approach because this technique is better for high-resolution imagery [69–71]. The object-based image analysis approach involves two interrelated stages: (1) segmentation and (2) classification [72,73]. The segmentation step aims to group pixels into "superpixels" or objects by minimizing variability within objects and maximizing it between objects [42]. Objects were made using the segment mean shift algorithm in ArcGIS Pro, which is a non-parametric approach to grouping pixels based on the Euclidean distance within spectral space [74–76]. Here, we selected input parameters including spectral and spatial details of 20 and a minimum region size of 10 by a trial-and-error process.

A random forest (RF) classifier was used to estimate the expansion and contraction of land cover classes over time with balanced training samples at the restoration and reference sites because previous research using this method suggested that RF classification consistently provides greater accuracy between images, which is ideal for multi-temporal image analysis [77]. Briefly, RF classification is a non-parametric machine learning algorithm that classifies land cover by establishing a set number of decision trees and bootstrap aggregation to generate many tree sequences and each tree sequence "votes" a classification label for a segmentation [78,79]. Each segment is given the classification label with the most "votes". We opted for the default RF parameters, including a maximum depth of tree of 10, a maximum number of trees in a forest of 20 and an out-of-bag error of 0.01, because previous research has demonstrated that these values produce optimal results [80,81]. The RF algorithm was conducted in Orfeo Toolbox (OTB) open-source software [82].

### 2.5. Accuracy Assessment

Between 456 and 779 validation points per class were randomly selected to assess the accuracy of each land cover class (Table 2). Due to restoration actions causing dynamic land cover changes throughout the timeline of this project, a new suite of training and validation samples was generated every six months. Confusion matrices were used to quantify class-specific accuracies via users' and producers' errors and two combined accuracy metrics for each map with overall accuracy (OA). For this study, we followed recommendations by Stehman and Foody [57] and weighted OA by the total area of each land cover class, which accounts for biases associated with class proportions within a study area.

### 2.6. Area Change Analysis

We conducted two area change analyses, including: (1) net change in each land cover class between 2017 spring (0 months) and 2021 spring (48 months) and (2) the effect of

time throughout the restoration trajectory on each land cover class. The net change in land cover area was calculated with GIS analysis by subtracting the distribution of each land cover map from 2017 spring (0 months) and 2021 spring (48 months). The effects of time (scale continuous explanatory variable) on the area of each land cover class (scale continuous response variable) at Yandina Creek wetlands were measured with generalized linear models (GLMs) in the R statistical framework [83]. GLMs were fitted with Gaussian distributions because all variables were continuous, and the distribution of residuals was normal. Significant effects between the area of each land cover class and time were plotted as regressions with R packages ggplot2 and ggpubr [84,85].

## 3. Results

### 3.1. Land Cover Classification

We generated a time-series of nine bi-annual land cover maps for Yandina Creek wetlands between 2017 and 2021 at an interval of approximately six months. Each map classified the distribution and coverage of nine land cover classes (mangroves, mangrove fern, saltmarsh, mixed she-oak and paperbark, mixed shrubs and grasses, common reed, vegetation dieback, exposed soils, tidal water) and shadows (Figure 3). The land cover maps recorded little variation and high area adjusted OA between 91% and 95% (Table 2).

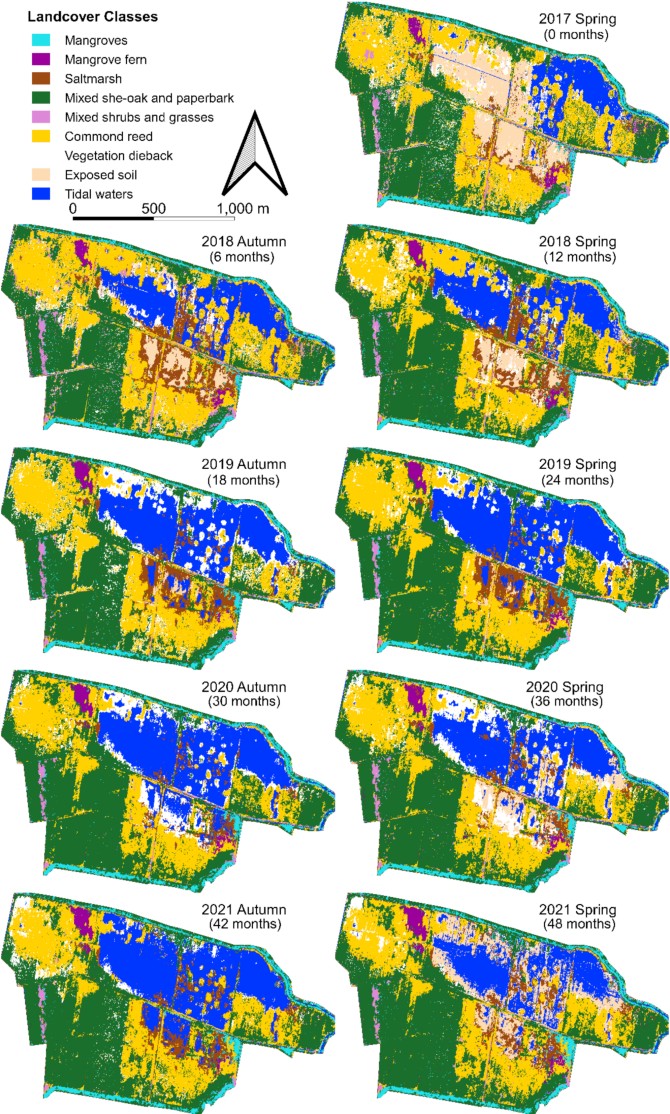

**Figure 3.** Bi-annual changes in the distribution and cover of land cover classes at Yandina Creek wetland between spring 2017 and 2021.

### 3.2. Class-Specific Classification

Users' and producers' accuracies were consistently high for all nine land cover classes (typically between 0.77 and 1 for both users' and producers' accuracy), and this was consistent over time. (Figure 4 and Table S3). Distinct vegetation communities, including mixed she-oak and paperbark, mangroves, mangrove fern and common reed, were characterized by consistently high users' and producers' accuracies through time, which were higher than spectrally similar classes such as saltmarsh and mixed shrubs and grasses (Figure 4 and Table S3). The users' and producers' accuracy of tidal water classifications also remained consistently high but did regress in accuracy as time progressed (Figure 4 and Table S3). Accuracies for exposed soil and vegetation dieback were mixed, with high variability in users' and producers' accuracies recorded (Figure 4 and Table S3).

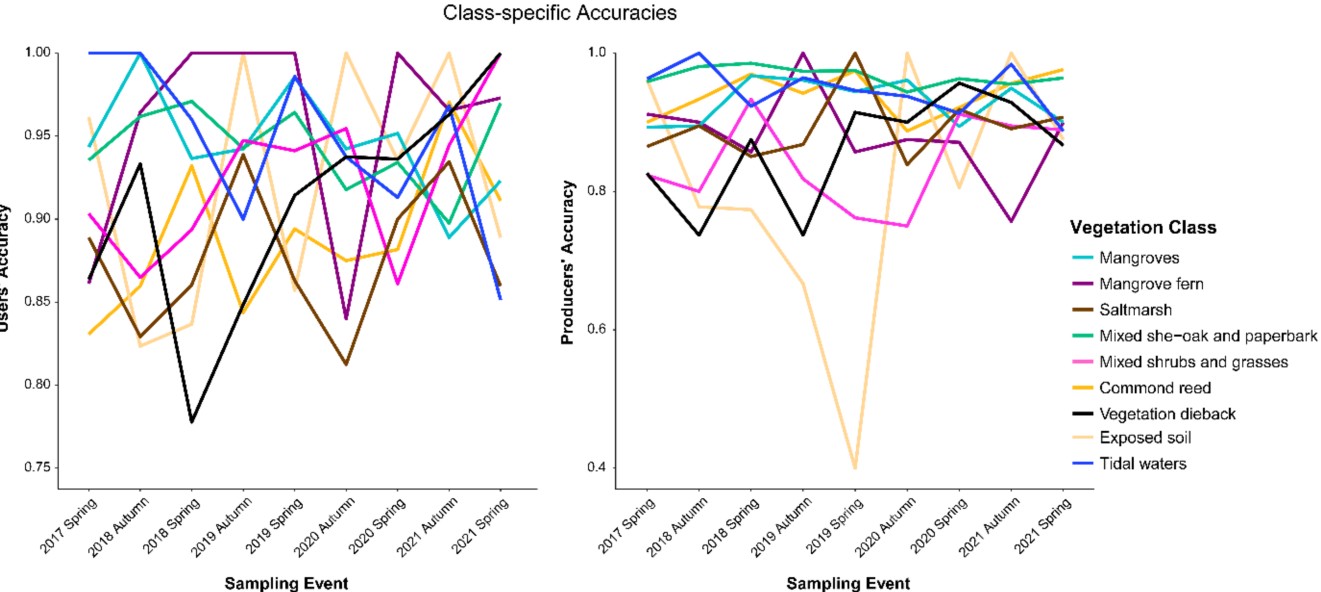

**Figure 4.** Accuracies of users' and producers' for the nine land cover classes on each of the nine maps produced between spring 2017 and 2021.

### 3.3. Restoration Effects on the Distribution and Coverage of Land Cover Classes

The distribution and cover of land cover classes at the Yandina Creek wetlands changed considerably between spring 2017 and 2021. Land cover classes that increased by the largest area included mixed she-oak and paperbark communities (an increase of 10 ha) and tidal water (an increase of 8.93 ha) (Figure 5). The extent of marine plant communities at the restoration site was also augmented between spring 2017 and 2021, with increases of in the area of saltmarsh (4.31 ha), mangrove fern (1.01 ha) and mangroves (0.91 ha). In contrast, there was a considerable reduction in the area of common reed (9.83 ha) and vegetation dieback (9.77 ha), and smaller reductions in the area of mixed shrubs and grasses (3.37 ha) and exposed soil (2.2 ha).

There was considerable spatial variation in the locations where change took place for each land cover class (Figure 6). Mixed she-oak and paperbark communities simply expanded at locations where these vegetation types were already established. The area of tidal waters expanded, and did so considerably in locations where obstructions to inundation (i.e., tidal gates) had been removed. Saltmarsh, mangrove fern and mangrove communities also expanded at locations where these vegetation types were already established (Figures 3 and 6). This occurred in areas that experienced tidal inundation, and also for saltmarsh in areas where tidal ingress was restricted. By contrast, some areas of common reed and vegetation dieback were replaced with either marine plant communities or exposed soil, and a large area of exposed soil was replaced with tidal waters (Figures 3 and 6).

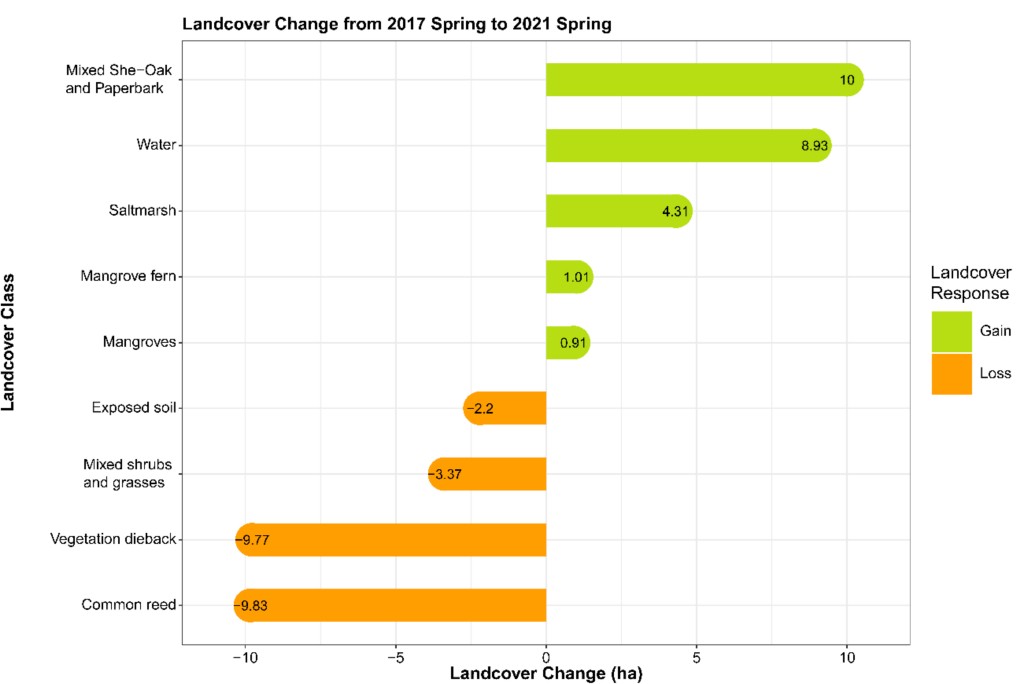

**Figure 5.** Changes in the area of each of the nine land cover classes between spring 2017 and 2021.

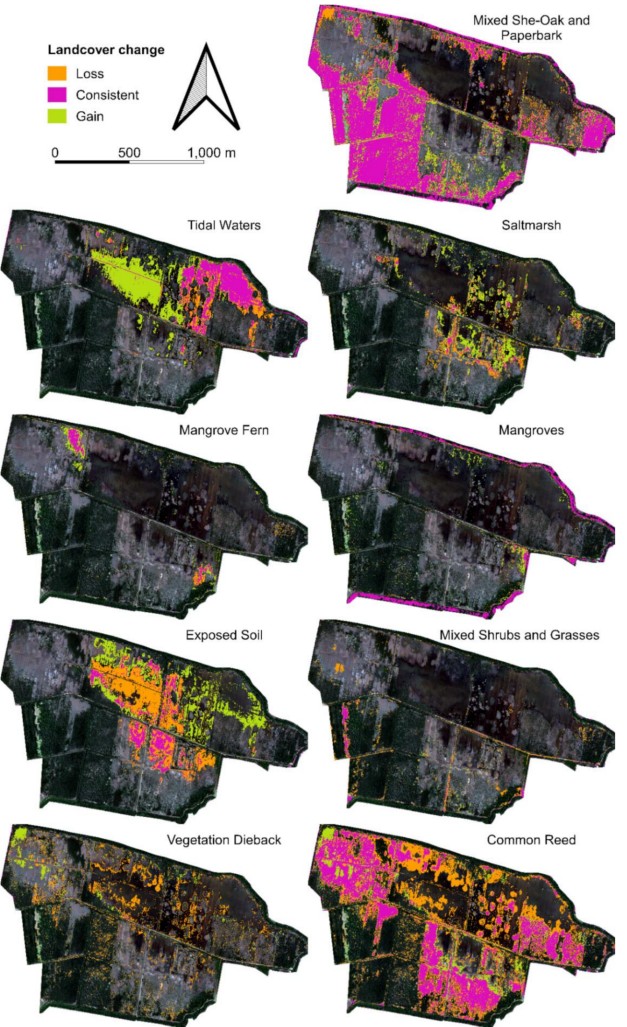

**Figure 6.** Spatial distribution of changes in the nine land cover classes between spring 2017 and 2021.

### 3.4. Trajectories of Change in the Area of Each Land Cover Classes between Spring 2017 and 2021

The area of three land cover classes, including mangroves, mangrove fern and mixed she-oak and paperbark communities, increased over time (Figure 7, Table S4). By contrast, the area of mixed shrubs and grasses and common reed declined over time (Figure 7, Table S4). The area of saltmarsh, vegetation dieback, exposed soil and tidal water did not increase or decrease significantly over time, despite clear gains and losses in area of each land cover class (see Figures 3 and 6) because the direction of changes varied over time (Table S4).

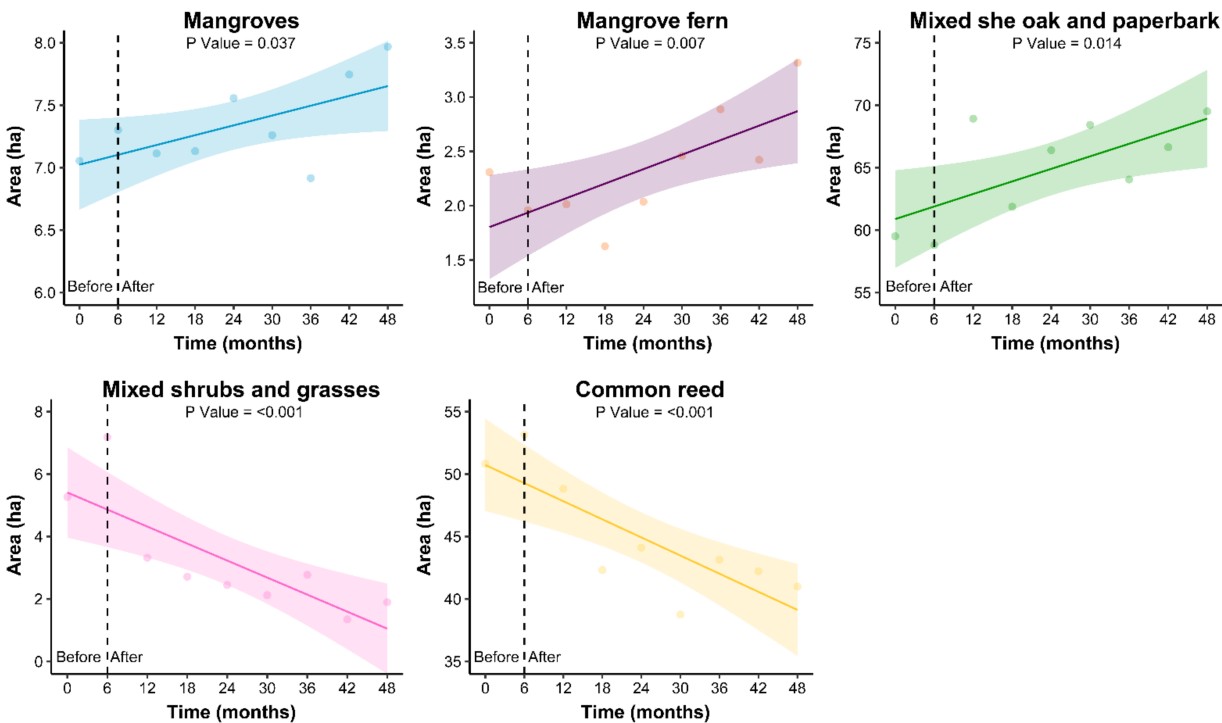

**Figure 7.** Generalized linear models (GLMs) illustrating changes in the area of wetland vegetation classes over a period of 48 months, between spring 2017 and 2021. Restoration commenced with the removal of tidal obstructions 6 months after the start of this monitoring program.

## 4. Discussion

This study developed and tested an accurate approach for monitoring the responses of coastal vegetation communities to wetland restoration. The overall accuracies of our analyses were higher than those reported in similar studies, which have used lower-resolution sensors, pixel-based classifications and/or parametric classifiers [25,86–88]. The combination of pan-sharpened Worldview-2 imagery, object-based image analysis and random forest classification were responsible for the high overall accuracies that were sustained over time. This is because pan-sharpened, high-resolution imagery compliments object-based image analysis more effectively than lower-resolution sensors or pixel-based approaches by generating more detailed, and spectrally and structurally distinct, objects that reduce pixel mixing errors [36,42,89]. Furthermore, random forest classifiers are able to better integrate detailed and data complex vectors, similar to the segments our approach generates, in comparison to other non-parametric classifiers such as support vector machines [39]. While community-level classifications of coastal vegetation communities have been achieved with freely accessible sensors, such as Sentinel-2, these yielded accuracies that were lower than our approach with the pay-for-service Worldview-2 sensor, which is an important consideration for wetland restoration projects that are limited by funding constraints [90].

A caveat of our study was classifying to community level rather than species level. This likely limits the capacity of our approach to inform adaptive management decisions for focal species and for estimating precise values associated with the delivery of ecosystem

services. For example, different mangrove species vary in the timing, location and scale of their responses to restoration actions, and this can impact on structure changes and biomass accumulation in mangrove ecosystems [47]. Drone-based approaches for acquiring aerial imagery can be used to produce highly accurate (90%) maps of the species composition of estuarine vegetation communities, but only at small spatial scales [71,91]. To map vegetation communities at the spatial and temporal scales of our study with drones, researchers would need multiple days of pre-flight planning and several weeks of drone flights, as these approaches are limited by small battery life, weather conditions and operating standards (e.g., no-fly areas) [59,71]. Many of these issues are manageable and cost-effective for short-term surveys of small study areas but become infeasible when scaled up to larger areas and timeframes. In contrast, our sensor-based approach is a viable option for monitoring coastal wetland restoration projects that require accurate and frequent community-level classifications over large spatial extents.

Our approach maintained consistent accuracy values over time by using a diverse range of structural and spectral datasets and multi-temporal mapping techniques [68,92]. Incorporating DEMs and CHMs was vital to separate spectrally similar communities, such as mangroves and mixed she-oak and paperbark, which differed in terms of their structure and elevation [93–96]. LiDAR-borne DEMs or CHMs are, however, constrained by limited temporal availability, and thus only DEMs and CHMs from 2018 were used to inform every land cover map, which impeded upon the capabilities of our approach to detect changes in land cover communities over time. Drone-based approaches provide better utility because they can generate highly accurate DSMs through structure from motion, which is a reason for future approaches to consider drone-based imagery [58,59]. Misregistration errors between the same features over time are common drawbacks of temporal object-based image analysis, and we accounted for these errors by including the previous land cover map to inform classifications [68,92]. In studies detecting changes to land cover classes, some vegetation communities often represent broad floristic and structural properties that vary spatially around a site because of their proximity and connectivity to nearby restoration actions [25,97]. We found these errors in some more dynamic classes such as dying vegetation, mangrove fern, mixed shrubs and grasses and saltmarsh, as they were all heavily influenced by the effects of restoration over time and expressed different spectral properties based on their distribution to restoration actions. Here, we recommend including spatial datasets which are increasingly used as ancillary datasets for differentiating between spectrally and structurally similar classes.

Our approach measured increases in the area of many threatened coastal wetland communities, including mixed she-oak and paperbark, mangroves and saltmarsh communities and losses in the coverage of dying vegetation, mixed shrubs and grasses and common reed monocultures, which are results that are difficult to measure with moderate or coarse resolution imagery [98,99]. These types of dynamic shifts over large areas are often not fully captured by more accurate and detailed, but less frequent, field surveys [21]. The changes we report here suggest that the successional effects of restoration have commenced for some coastal vegetation communities, but also that the magnitude of responses to restoration actions varies considerably among vegetation types. Differences in the responses of wetland communities are likely affected by the dispersion capabilities and growth attributes of focal species and the availability and proximity of remnant communities to serve as source populations for seeds and propagules [100,101]. For example, saltmarshes likely responded faster and at a large spatial scale than mangroves because saltmarshes are capable of asexual reproduction and rapid expansion through their stolons and rhizomes, and also because mangroves have slow growth rates when passive restoration actions are chosen over active techniques as propagule availability and accessibility can be limited [102–104]. Despite effectively identifying trends in the coverage of wetland communities, our approach was limited by temporal fluctuations in class-specific area estimates over time. This variance between land cover classes is common with satellite-based sensors due to the impacts of horizontal co-registration errors and variation in the off-nadir angle of the sensor and the

sun, which can contribute to varying aspects and object sizes when quantifying land cover types [105]. Emerging automated geo/co-registration frameworks and current rational polynomial coefficient models and orthoengine software exist to handle misalignment errors associated with co-registration and off-nadir errors with very high- and high-resolution satellite imagery [106,107]. Therefore, future applications of this approach should consider that accounting for co-registration and off-nadir errors might provide more robust estimates of temporal changes in the area of coastal wetland communities.

Future methods should build on our approach by quantifying species-level responses to restoration actions, which would provide more precise adaptive management strategies and evaluations of ecosystem service regeneration. Complex deep learning approaches can identify to species level, but currently require high computation power, large budgets, and thousands of training datasets, and this impacts on the feasibility of these approaches for many applications [33,41]. It is expected the viability of deep learning systems will become greater over time as advanced computer hardware becomes more affordable. Drones are emerging as useful tools for mapping coastal wetland communities because they provide very high-resolution orthomosaics (<2 cm), generate DSMs through LiDAR or structure from motion and are quick to use across multiple repeated deployments. The ability of drones to complete frequent and large-scale mapping of coastal wetlands will continue to improve as battery technology progresses [91,108–110]. Satellite sensors are also likely to improve with enhancements to image resolution, increases in spectral range and faster revisitation rates, which improves temporal coverage. This is evident in recently deployed satellite constellations such as the Planet SuperDove with eight multispectral bands, 3 m resolution and access to daily and even sub-daily imagery, which is highly valuable for real-time monitoring of water bodies [111]. Furthermore, the use of cloud-based computing systems (e.g., Amazon, Google, and Openeo by vito) can complement the advancement in technology and computer-intensive methods such as very high-resolution images and deep learning image analysis. These approaches may even reduce the costs associated with classification whilst handling large datasets [13,32,112–114]. The potential for approaches to extract accurate species-level identifications provides exciting opportunities for researchers to monitor the changes to restoring coastal wetlands in the future. This will be highly valuable for informing species-specific adaptive management strategies and precise evaluations on the regeneration of ecosystem services throughout the timeline of restoring a coastal wetland.

## 5. Conclusions

We used a high-resolution sensor, object-based image analysis and a non-parametric machine learning classifier to detect and describe spatial and temporal changes in the area and distribution of coastal vegetation communities in response to wetland restoration actions. Our results demonstrate that this approach can provide an accurate and replicable tool for monitoring changes in restoring coastal wetland communities. This high-resolution, sensor-based approach might be particularly useful for monitoring responses of vegetation communities to coastal restoration initiatives because it can be readily employed to describe changes in area and composition of focal vegetation communities at comparatively large scales (100 s of hectares across multiple years), aligning with both the area and timeframe of adaptive management decisions.

**Supplementary Materials:** The following supporting information can be downloaded at: https://www.mdpi.com/article/10.3390/rs14184559/s1, Table S1: Summary of datasets, the platform they were derived from, the dates used, their respective spatial resolution and reference details, Table TAB;: Summary of the specifications of the Worldview-2 sensor, Table S3: Confusion matrices that demonstrate the agreement and precision of validation points compared to classified outputs for each land cover map, Table S4: Generalised Linear Models (GLMs) summarising changes in the area of wetland vegetation classes over a period of 48 months, between Spring 2017 and 2021 over-time. Arrows illustrate the direction of change, as an increase in area (↑), a decrease in area (↓) or no consistent pattern of change (-).

**Author Contributions:** Study conceptualization and design, A.J.R., J.X.L. and A.D.O.; data collection, A.J.R., J.X.L., B.B.E. and H.P.B.; method development, A.J.R. and J.X.L.; data curation and analysis, A.J.R., J.X.L., H.P.B., B.L.G. and C.J.H.; writing, review and editing, A.J.R., J.X.L., H.P.B., B.B.E., B.L.G., C.J.H. and A.D.O. All authors have read and agreed to the published version of the manuscript.

**Funding:** This research was supported by funding received from Unitywater (ADO, BLG).

**Institutional Review Board Statement:** Not applicable.

**Informed Consent Statement:** Not applicable.

**Data Availability Statement:** Not applicable.

**Acknowledgments:** We would like to acknowledge Unitywater and the University of the Sunshine Coast for supporting this research.

**Conflicts of Interest:** The authors declare no conflict of interest.

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
