# Peer review of "Watching the Saltmarsh Grow: A High-Resolution Remote Sensing Approach to Quantify the Effects of Wetland Restoration"

_remotesensing, doi:10.3390/rs14184559_

Round 1

Reviewer 1 Report

Dear Authors,

After reading your paper, I have the following comments:

1. Why are the validation data points around 1/3 of the training data points? Please make it around 50/50, which will make the manuscript more strong.

2. Please include a separate analysis considering only raw data (without random selection). Hence, discuss the significance of the random selection process. 

3. The discussion part is very weak based on the current version. The authors need to explain each result's significance and signal the positive or negative relationship between the available literature and recent work.

4. The authors need to clearly state and validate the uniqueness of the current work from previous literature.

Author Response

Comment 1: Why are the validation data points around 1/3 of the training data points? Please make it around 50/50, which will make the manuscript more strong.

Response: We thank Reviewer 1 for the suggestion. We used a 2/3 for training and 1/3 testing ratio based on recent publications (Lyons et al., 2018). We have now added this into the methods section to clarify this and included this citation. Please see line edits and tracked changes for lines 140-141.

Comment 2: Please include a separate analysis considering only raw data (without random selection). Hence, discuss the significance of the random selection process. 

We followed (Stehman and Foody, 2019), which outlined the approaches of adopting simple random sampling design. We have included this with the citation to highlight this.  Please see line edits and tracked changes for lines 141-142.

Comment 3: The discussion part is very weak based on the current version. The authors need to explain each result's significance and signal the positive or negative relationship between the available literature and recent work.

Thank you for the comment reviewer 1, we have explored the positive and negative relationships from our findings throughout our discussion section. We have also further expanded the discussinoa nd conclusion section by exploring the potential of cloud-based approaches: lines 407 – 411.

Comment 4: The authors need to clearly state and validate the uniqueness of the current work from previous literature.

We thank reviewer 1 for this comment, we have expanded the uniqueness of our approach to also discuss the implications of mapping degradation and cloud-based approaches in the introductions, whilst also emphasising the need for the measuring of targeted ecosystems in restoration. See lines 46-47 and 51-52.

Thank you for your suggestions throughout. Best wishes. 

LYONS, M. B., KEITH, D. A., PHINN, S. R., MASON, T. J. & ELITH, J. 2018. A comparison of resampling methods for remote sensing classification and accuracy assessment. Remote Sensing of Environment, 208, 145-153.

STEHMAN, S. V. & FOODY, G. M. 2019. Key issues in rigorous accuracy assessment of land cover products. Remote Sensing of Environment, 231, 111199.

Reviewer 2 Report

Using visual effects and novel presentation techniques will improve interests of readers and it will provide more visual and citation. 
Using " We and Our, in scientific article is nor recommended.
Very good referencing (more than 20% most recent core articles)

Author Response

Comment 1: Using visual effects and novel presentation techniques will improve interests of readers and it will provide more visual and citation. 

Response: We thank reviewer 2, and agree with this feedback and have modified our graphical abstract to be more visually appealing.

Comment 2: Using " We and Our, in scientific article is nor recommended.

Response: We disagree with this comment, use of the first person of active voice is recommended and commonly used when writing scientific articles – see rule 7 of (Hotaling, 2020)

Comment 3: Very good referencing (more than 20% most recent core articles)

Response: We thank reviewer 2 for this comment. We aim to use not only recent articles, but also relevant ones.

HOTALING, S. 2020. Simple rules for concise scientific writing. Limnology and Oceanography Letters, 5, 379-383.

Reviewer 3 Report

Dear Author(s)

I agree, the manuscript concerns very crucial issue on restoration of wetlands. Due to world-wide politics and climate changes restoration of wetlands seems to be the important research area. However I have a few major concerns the author(s) should answer and include them for improving the manuscript.

1. Introduction (page 2 of 24) should concern information on degradation as well. If you study the changes at wetlands, you ought to mention not only on restoration process but also on degradation one. The both processess are interrelated. Thus it will be suitable to mention on some applications of remote sensing data for investigating degradation of wetlands (see paper below)

Zou, Z.; DeVries, B.; Huang, C.; Lang, M.W.; Thielke, S.; McCarty, G.W.; Robertson, A.G.; Knopf, J.; Wells, A.F.; Macander, M.J.; Du, L. Characterizing Wetland Inundation and Vegetation Dynamics in the Arctic Coastal Plain Using Recent Satellite Data and Field Photos. Remote Sens. 2021, 13, 1492. https://doi.org/10.3390/rs13081492

Šimanauskienė R., Linkevičienė R., Bartold M., Dąbrowska‐Zielińska K., Slavinskienė G., Veteikis D., Taminskas J., 2019, Peatland degradation: the relationship between raised bog hydrology and NDVI, Ecohydrology 2019, e2159. doi:10.1002/eco.2159

Alonso, Alice & Muñoz-Carpena, R. & Kennedy, R.E. & Murcia, Carolina. (2016). Wetland landscape spatio-temporal degradation dynamics using the new google earth engine cloud-based platform: Opportunities for non-specialists in remote sensing. 59. 1333-1344. 10.13031/trans.59.11608.

Ján Feranec, Marcel Šúri, Ján Ot'ahel', Tomáš Cebecauer, Ján Kolář, Tomáš Soukup, Dagmar Zdeňková, Jiří Waszmuth, Vasile Vâjdea, Anca-Marina Vîjdea, Constantin Nitica,
Inventory of major landscape changes in the Czech Republic, Hungary, Romania and Slovak Republic 1970s – 1990s, International Journal of Applied Earth Observation and Geoinformation, Volume 2, Issue 2,
2000, Pages 129-139, ISSN 1569-8432, https://doi.org/10.1016/S0303-2434(00)85006-0. 

2. Introdution (page 2 of 24) I think the remote sensing-based and operational systems for mapping wetlands for their protection should be also mentioned. For example there are commonly known GlobWetland (GW-A Toolbox for Africa) or implemented in Poland system (POLWET) are good examples to rise awareness about application of remote sensing data for monitoring wetlands. Please find paper to be presented below concerning satellite-based products supporting managing wetlands.

Dabrowska-Zielinska K., Bartold M., Gurdak R., 2016, POLWET – System for new space-based products for wetlands under RAMSAR Convention, Geoinformation Issues, Vol. 8, No. 1(8), pp. 25-35. doi:10.34867/gi.2016.3 http://bc.igik.edu.pl/Content/608/PDF/GI%20Vol.%208%201(8)_4.pdf

Shuai Xiaoying, Qian Huanyan, Design of Wetland Monitoring System Based on the Internet of Things, Procedia Environmental Sciences, Volume 10, Part B, 2011, Pages 1046-1051, ISSN 1878-0296, https://doi.org/10.1016/j.proenv.2011.09.167.

B. Wolf, "GlobWetland II: Wetland mapping in North Africa," 2011 GEOSS Workshop XLI, 2011, pp. 1-40, doi: 10.1109/GEOSS-XLI.2011.6047973. 

3. Materials and methods (5 of 24)
My general question is: why do you classify each of eight(nine) wv-2 satellite images and next investigate the changes? I mean, did you try to generate layer stack concerning all nine satellite-images firstly and then find pixel-by-pixel (or object-by-object) the changes directly with rf method?
Current remote-sensing based methods are mostly implemented in cloud-computing systems (gee, amazon, openeo by vito). I realize world-view-2 are commercial data and require proportionately high budget for collectin' them. However, machine learning methods are offered at cloud services at quite low price and sometimes free of charge as well.

4. Discussion:

I can see you obtained very high classification results at least 90% of accuracy. However you require high-priced wv-2 satellite data for achieving these results. What about low or medium satellite data (Sentinel-2)? Could you add sentences on possibility of application them for? If the results were with at least 80% accuracy applying lower resolution and free of charge(!) satellite data, then you could accept them for your study area?

5. Results and discussion:

I think the author(s) should mention on possibility of application their approach at operational and national scale. Google Earth Engine, Amazonw Web Service etc. are powerful cloud-computing systems that offer remote-sensing tools for mapping the environment as well.

Best wishes

Author Response

Reviewer 3

Comment 1. Introduction (page 2 of 24) should concern information on degradation as well. If you study the changes at wetlands, you ought to mention not only on restoration process but also on degradation one. The both processess are interrelated. Thus it will be suitable to mention on some applications of remote sensing data for investigating degradation of wetlands (see paper below).

Zou, Z.; DeVries, B.; Huang, C.; Lang, M.W.; Thielke, S.; McCarty, G.W.; Robertson, A.G.; Knopf, J.; Wells, A.F.; Macander, M.J.; Du, L. Characterizing Wetland Inundation and Vegetation Dynamics in the Arctic Coastal Plain Using Recent Satellite Data and Field Photos. Remote Sens. 2021, 13, 1492. https://doi.org/10.3390/rs13081492

Šimanauskienė R., Linkevičienė R., Bartold M., Dąbrowska‐Zielińska K., Slavinskienė G., Veteikis D., Taminskas J., 2019, Peatland degradation: the relationship between raised bog hydrology and NDVI, Ecohydrology 2019, e2159. doi:10.1002/eco.2159

Alonso, Alice & Muñoz-Carpena, R. & Kennedy, R.E. & Murcia, Carolina. (2016). Wetland landscape spatio-temporal degradation dynamics using the new google earth engine cloud-based platform: Opportunities for non-specialists in remote sensing. 59. 1333-1344. 10.13031/trans.59.11608.

Ján Feranec, Marcel Šúri, Ján Ot'ahel', Tomáš Cebecauer, Ján Kolář, Tomáš Soukup, Dagmar Zdeňková, Jiří Waszmuth, Vasile Vâjdea, Anca-Marina Vîjdea, Constantin Nitica,
Inventory of major landscape changes in the Czech Republic, Hungary, Romania and Slovak Republic 1970s – 1990s, International Journal of Applied Earth Observation and Geoinformation, Volume 2, Issue 2,
2000, Pages 129-139, ISSN 1569-8432, https://doi.org/10.1016/S0303-2434(00)85006-0. 

Response: We thank reviewer 3 and we agree with the comment as this now broadens the scope of this paper. The references are fantastic additions. We have now modified the introduction and added the suggested references. See line edits and tracked changes at lines 46-47 and 51 and 52.

Comment 2. Introdution (page 2 of 24) I think the remote sensing-based and operational systems for mapping wetlands for their protection should be also mentioned. For example there are commonly known GlobWetland (GW-A Toolbox for Africa) or implemented in Poland system (POLWET) are good examples to rise awareness about application of remote sensing data for monitoring wetlands. Please find paper to be presented below concerning satellite-based products supporting managing wetlands.

Dabrowska-Zielinska K., Bartold M., Gurdak R., 2016, POLWET – System for new space-based products for wetlands under RAMSAR Convention, Geoinformation Issues, Vol. 8, No. 1(8), pp. 25-35. doi:10.34867/gi.2016.3 http://bc.igik.edu.pl/Content/608/PDF/GI%20Vol.%208%201(8)_4.pdf

Shuai Xiaoying, Qian Huanyan, Design of Wetland Monitoring System Based on the Internet of Things, Procedia Environmental Sciences, Volume 10, Part B, 2011, Pages 1046-1051, ISSN 1878-0296, https://doi.org/10.1016/j.proenv.2011.09.167.

B. Wolf, "GlobWetland II: Wetland mapping in North Africa," 2011 GEOSS Workshop XLI, 2011, pp. 1-40, doi: 10.1109/GEOSS-XLI.2011.6047973. 

Response: We thank reviewer 3 and we agree with the comment. We have made changes to page 2 of the introduction and have included all recommended citations, thank you for those. See line edits and tracked changes at lines 70-72.

Comment 3: My general question is: why do you classify each of eight(nine) wv-2 satellite images and next investigate the changes? I mean, did you try to generate layer stack concerning all nine satellite-images firstly and then find pixel-by-pixel (or object-by-object) the changes directly with rf method?
Current remote-sensing based methods are mostly implemented in cloud-computing systems (gee, amazon, openeo by vito). I realize world-view-2 are commercial data and require proportionately high budget for collectin' them. However, machine learning methods are offered at cloud services at quite low price and sometimes free of charge as well.

Thank you for the question, we agree that would be best and we are familiar with similar work (e.g. Hölbling et al, 2015, Anders et al 2013 & Chen et al, 2012).  However, our aim was to develop an approach that could be repeated in the future even with different datasets (e.g. drone imagery or other high-res satellite imagery).

Also, We thank you for second comment here and we have updated the discussion discussing the value of online services such as amazon etc. In the future, we could consider the use of cloud based sources to process high resolution data using deep learning methods. See line edits and tracked changes: 407-411

Comment 4: Discussion: I can see you obtained very high classification results at least 90% of accuracy. However you require high-priced wv-2 satellite data for achieving these results. What about low or medium satellite data (Sentinel-2)? Could you add sentences on possibility of application them for? If the results were with at least 80% accuracy applying lower resolution and free of charge(!) satellite data, then you could accept them for your study area?

Response: Thank you for the comment, we have mentioned the us of sentinel 2 with similar approaches, so in this case, given we had the resources we decided to use Worldview-2. We required finer resolution to differentiate between classes even though it is more expensive. Please lines: 329 - 335

Comment 5: Results and discussion: I think the author(s) should mention on possibility of application their approach at operational and national scale. Google Earth Engine, Amazonw Web Service etc. are powerful cloud-computing systems that offer remote-sensing tools for mapping the environment as well.

Dear reviewer 3, thank you fo your comment, we have addressed this in the discussion and conclusion section in lines 407-411. Your comments have much appreciated.

Best wishes.

Round 2

Reviewer 3 Report

Thank your for manuscript. I accept in present form.